# Tor Hidden Services: A Systematic Literature Review

Diana L. Huete Trujillo [†] and Antonio Ruiz-Martínez *,[†]

Department of Information and Communications Engineering, University of Murcia, 30100 Murcia, Spain; dianalissette.huete@um.es
* Correspondence: arm@um.es
† These authors contributed equally to this work.

**Abstract:** Anonymous communications networks were created to protect the privacy of communications, preventing censorship and traffic analysis. The most famous anonymous communication network is Tor. This anonymous communication network provides some interesting features. Among them, we can mention that Tor can hide a user's IP address when accessing to a service such as the Web, and it also supports Tor hidden services (THS) (now named onion services) as a mechanism to conceal the server's IP address, used mainly to provide anonymity to websites. THS is an important research field in Tor. However, there is a lack of reviews that sum up the main findings and research challenges. In this article, we present a systematic literature review that aims to offer a comprehensive overview of the research made on THS by presenting the state-of-the-art and the different research challenges to be addressed. This review has been developed from a selection of 57 articles and presents main findings and advances regarding Tor hidden services, limitations found, and future issues to be investigated.

**Keywords:** Tor; hidden services; onion services; systematic literature review; survey





## 1. Introduction

The Onion Router (Tor) [1] is currently the broadest low-latency anonymous communication network and is used every day by many people to protect their privacy and overcome censorship [2,3]. The common belief in the Tor project is that "[I]nternet users should private access to an uncensored web" [4].

The main function that provides Tor to any citizen is, through a set of nodes in the Tor network, to protect his/her anonymity by hiding his/her network IP address [1]. The security and anonymity of Tor have been deeply studied in many works [2,5], and although different types of attacks are possible [3,6], the research community is working toward providing solutions to the different attacks [5,7,8].

However, Tor goes beyond protecting a citizen's network identity. Since 2004, it has offered the possibility of protecting the locations of services through a mechanism that formerly was named Tor hidden services (HS) and that is currently named onion services (OS). Thus, anyone can provide services (*hidden services* or *onion services*), and the server's IP address cannot be learned by a client of the service.

These services can be clearly identified because their addresses are finished in .onion. According to Tor metrics, there are more than 150,000 of these hidden services [9]. Any hidden service requires that any citizen executes the rendezvous protocol to contact the service. This protocol guarantees that the citizen accesses the service without having knowledge of server's IP address. While the citizen is using Tor, the service does not know citizen's IP address. Thus, using this protocol both the citizens and server IP addresses are hidden from each other, and other entities cannot see that they are connected.

Since their introduction, the number of HS offered has exponentially increased [9]. Users use them to access services related to various topics, such as pornography, drugs, politics, security, and anonymity. Cybercriminals make use of HS to perform illegal

activities [10]. These kinds of activities are related to counterfeit credit cards, drugs, pornography, and weapons. However, although it is used for these activities, which are not desired, as Snowden said, "Tor is a critical technology, not just in terms of privacy protection, but in defense of our publication right" [11]. Tor is a key tool to circumvent censorship. Indeed, it is the most widely system used for this purpose.

The whole set of services that are accessed in this way with other services from other anonymous networks that follow the same approach forms what it is called the dark web [12]. This dark web has become famous in mass media because many illicit services have been offered through darknet markets [13] or cryptomarkets [14]. However, as mentioned, Tor aims to help circumvent censorship and support freedom of speech. Thus, it is used by journalists and dissidents; and to provide freedom of speech it is being used by human rights and whistle-blowing organisations such as Wikileaks and Goldballeaks [15]. It is also fundamental to a secure political freedom [16]. Furthermore, there is an important number of HS providing links to services of the surface Web—namely, to social networks, web content management, news, and adult content [12]. Thus, Tor is a neutral tool that, depending of the purposes we use it for, can be either good or bad [16].

As mentioned previously, due to its popularity and the services it offers, Tor has been broadly studied and many issues regarding it have been analysed. It has been analysed both from the user's point of view [17] and a technical point of view [5,12,15]. Technical studies are the most common analyses.

In a taxonomy presented by Saleh et al. [5], they classified Tor research topics into de-anonymisation, performance analysis and architectural improvements, and path selection. Within these topics, HS were included. But, in this analysis, they only covered four papers. These hidden services are key within the Tor network and we believe they deserve to be studied in detail. However, we have not found a survey that tried to sum up what the research community has learnt about these hidden services and to identify what research issues still need to be addressed or improved. Therefore, we decided to analyse the state-of-the-art of Tor hidden services. For this purpose, we decided to address this issue through a systematic literature review (SLR), wherein we have analysed THS from different points of view. Our methodology was based on SLR because it follows a well-defined methodology which makes sure that the research on a specific issue becomes exhaustive, fair, and repeatable [18].

In this paper, we present what we discovered after analysing the research on THS during the period of 2006 to 2019 by means of a SLR. Namely, we have analysed 57 papers, and this paper explains how we developed the survey, the questions we asked, and the answers we obtained regarding the state-of-the-art of THS.

The rest of this paper is structured as follows. In Section 2, we briefly present how THS works. Next, we present related work on Tor and HS in Section 3. In Section 4, we explain the methodology we followed to develop the SLR. Then, in Section 5, we present the results obtained and a discussion about them. Finally, we conclude the paper in Section 6.

## 2. Tor Hidden Services in a Nutshell

Tor provides anonymity for a receiver or a service provider through what is known as onion services or hidden services (HS) [1]. Next, we briefly explain the main features of Tor and how it works, and then we explain the main idea behind the communication with Tor hidden services.

### 2.1. Tor

Tor [1] is a distributed overlay network, composed of a set of nodes named onion routers (OR) or relays, to anonymize TCP-based applications. When a client wants access to a TCP-based service, such as the Web or instant messaging, he has to build a path to exchange information with the service.

In Figure 1, we show a connection between a client and a website. To create this connection, the client obtains a list of Tor nodes from the directory (step 1). From the list,

in most cases, he chooses three nodes for building the circuit: an entry guard (it could also be a bridge, which is a relay that is not publicly listed in the directory), a middle relay, and an exit relay.

The process of building the circuit is summed up next. This is an incremental process where the client, first, creates a circuit with the entry guard (step 2). For creating this circuit, a TLS connection between the client and the entry guard is established where they negotiate, using a Diffie–Hellman handshake, a key to exchanging information within the TLS connection. The information exchanged in the circuit is based on a data structure named *cell*, whose size is fixed at 512 bytes. Next, the client requests the entry guard to extend the circuit to the middle relay (step 3). In this process, the client negotiates a key with the middle relay in the same way he did it with the entry guard. Finally, the client extends the circuit to the third relay (step 4), the exit relay, which will be used to send the client's traffic to the website. The way the circuit is extended to include the exit relay is the same used to extend the circuit in the previous step. Thus, the client has established a path with three nodes, and with each node has negotiated an encryption (session) key. It is also important to point out that any relay in the path only knows its predecessor and successor, but no the rest of the relays in the circuit.

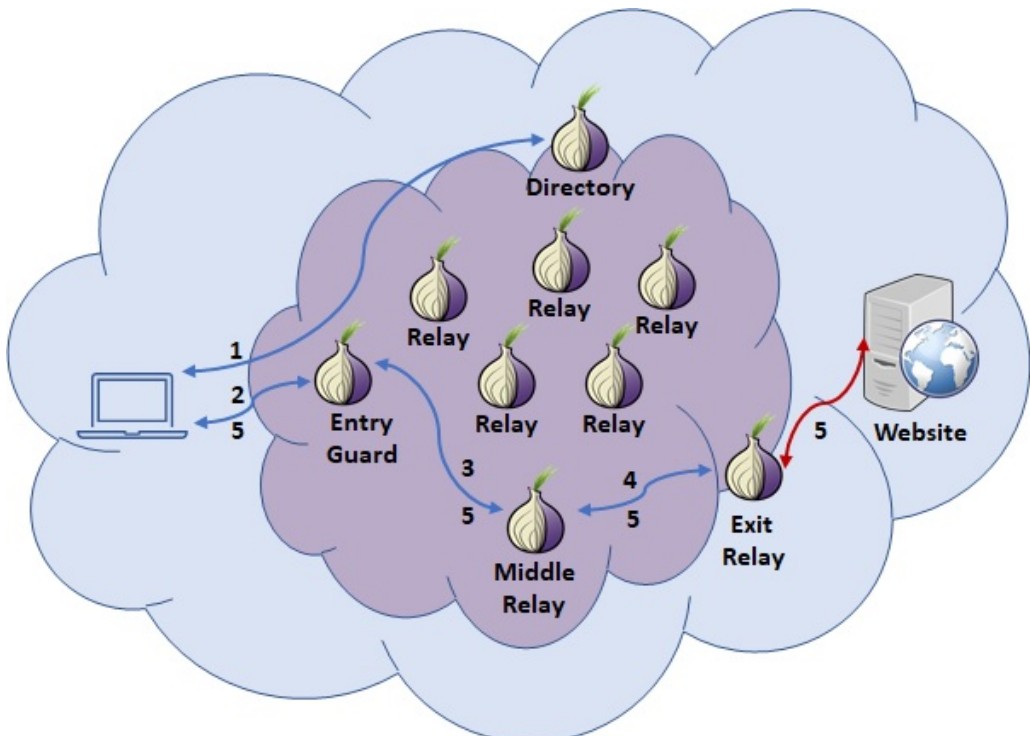

**Figure 1.** Tor circuit. Source: [19].

Once the circuit is built, the client can exchange information with the website (step 5). From the website's point of view, the connection comes from the exit relay—that is, who is sending the data received from the client through the circuit. These exchanges are sent through the path established by means of relay cells. In each cell, the client sends her data through relay cells. The data are within the payload, which is encrypted with the key negotiated with each relay. Namely, the payload is ciphered, first with the exit relay key. Next, the result is ciphered with the middle relay key, and finally, the result is ciphered with the guard relay key. As we can see, there are several layers of encryption, and in each relay, a layer of encryption is removed. This is the base of the onion ciphering. It is also important to point out that in each circuit we can multiplex many TCP streams. Thus, we could use it to connect to several websites or other services, such as instant messaging.

A more detailed description of Tor, its features, and the protocol can be found in [1].

## 2.2. Tor Hidden Services

For a user to communicate with a hidden service, a connection that is made of two circuits must be established: one from the user to a rendezvous point (RP) and one from the hidden service to the RP. Therefore, the path between the user and HS consists of six OR: three for the first circuit and three for the second. This access to the HS (dark web) requires twice as many nodes are required as those required to make a circuit to a site outside the Tor network (Surface web).

In Figure 2, we depict the process followed to establish a connection between a client and a HS/OS. This process could be divided into two phases. The first phase is the setup of the HS and its announcement (steps 1–3). Then, in the second phase, once the service is announced, some clients could be interested in it and will access it (steps 4–11).

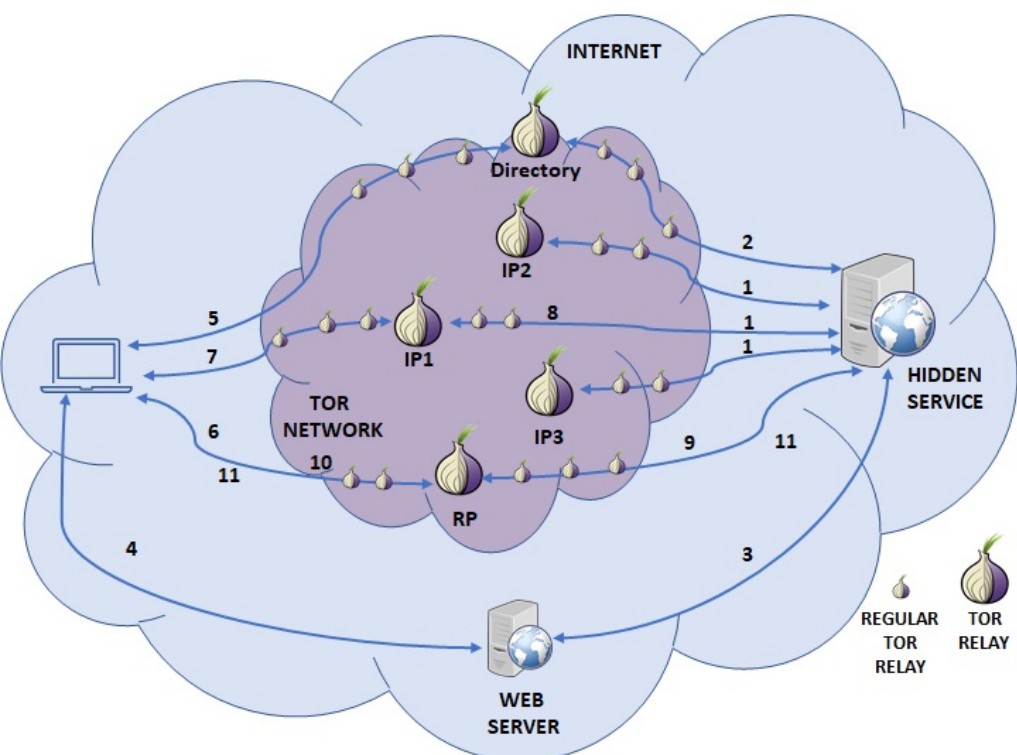

**Figure 2.** Tor hidden/onion service protocol. Source: [19].

To set up the HS, the service picks up some relays and establishes long-term circuits to them (step 1). With these circuits, the service asks them to become introduction points (IP) by sending them its public key. Next, the service creates an onion service descriptor (OSD) and uploads it to the directory (step 2), which is a distributed hash table. The OSD is signed by the service and contains its public key and a list with information about the introduction points chosen. Finally, the service publishes (step 3) its onion service address (an address finished in .onion, which in version 2 of the protocol has 16 characters and in version 3 has 56) to a (dark) web page, in a blog, by sending it via email, etc.

Once a user knows the onion address (step 4), he can start the process of accessing the HS. For this purpose, he creates a circuit to the directory and obtains the OSD corresponding to the onion address (step 5). He verifies the OSD using the public key encoded in the onion address. When successfully verified, he establishes a circuit to a relay, asking that it behave as a rendezvous point (step 6). He provides a one-time secret named a cookie to the RP. Next, he establishes a circuit to an IP of the service (step 7) and passes it the cookie and the RP address. Then, the IP forwards this information to the HS (step 8). The HS builds a circuit to the RP and sends the cookie (step 9). If the cookie matches with the cookie sent by the client, the RP notifies her that the connection was successfully established (step 10).

Finally, the connection between the client and HS is established and they can exchange information (step 11).

More details on hidden services, the directory server, and the rendezvous protocol can be found in Tor's specifications [20] and its design documentation [1].

To use the Tor network to conceal one's IP when accessing the surface web or for accessing HS (Dark web), users can install TorBrowser, which contains a customised version of Firefox, which connects to the Tor network and does not store browsing history; cookies are stored only for a session; it prevents browser fingerprinting; and includes some plug-ins to improve user's privacy when surfing the (dark) web, such as HTTPS Everywhere and NoScript. It is important to point out that to preserve our privacy on the web, it is not enough to hide our IP address; we must consider other mechanisms can be used to track us when surfing on the web such as cookies, Javascript, CSS, etc. [21,22].

## 3. Related Work

In this section we present some related work on Tor. Namely, we will focus on Tor hidden services and previous surveys.

In the literature, to the best of our knowledge, there is no a comprehensive survey focused directly on HS. Despite this fact, we can mention that there are several reviews of anonymous communication systems (ACS): two of them are focused on Tor, and there is anoter focused on the de-anonymisation of hidden services.

We can start the review of the different surveys on ACS with the work of Ren and Wu [23]. They reviewed the main techniques in the field of ACS, including Tor, and also included a quick review of hidden services. In their work, the main topics covered addressed security issues, and they concluded by exposing that the main vulnerability for a HS in Tor is the selection of the first and last node in the communication path.

Next, we can consider Alsabah and Goldberg's survey research on Tor [24]. They examined its design, identified weaknesses and deficiencies, and presented a classification system for the directions that the research is taking in these areas. As for HS, their greatest findings were again in security matters. In this work, they concluded by stating that improvements are needed in the design of THS. They mainly exposed the problems with malicious services directories and problems with services that accept "anonymous" payments with Bitcoin. They also provided a walk-through of documents that mention the ease with which a command and control (C&C) server may be protected behind an HS.

Alidoost Nia and Ruiz-Martínez [2] also published, in 2017, a systematic review of the literature on anonymous communication systems. They conducted their review based on the collection of information in seven academic databases and selected 203 papers for analysis. Of these papers, nine covered issues related to hidden services: aspects related to security, main attacks, proposals to strengthen security, analysis, and measurements of anonymity were addressed. The main topic was the de-anonymisation of hidden services. Finally, from this SLR, from all the papers analysed, the paper with the most references was a paper related to HS [25], which focused on trawling for HS. This paper showed the research interest in services of this kind.

More recently, Saleh et al. [5] published a review focused on Tor. This review classified the collected articles into three main groups: de-anonymisation, path selection, and improvement and performance analysis. Of all the articles that analysed Tor, only 9% were related to hidden services, and most of these focused on de-anonymisation, indicating that relays and traffic are the most susceptible factors.

As a last general review on Tor, we can mention Basyoni et al.'s [3] review. They analysed traffic analysis attacks on Tor and evaluated how practically these attacks can be performed in real-time in the Tor network. They pointed out that many of the de-anonymisation attacks on HS, based on Tor's threat model, assume that there are one or multiple malicious Tor relays. This model is also considered in flow watermarking attacks, where the malicious client injects the watermark into the flow communication, and then it can be detected by the corrupted entry guards of the hidden service and break anonymity.

The analysis provided a classification of the nine attacks analysed, but they did not cover HS in detail.

Finally, Nepal et al.'s survey [26] is the only paper that presents a review on a particular issue of HS: attacks schemes for revealing hidden servers' identities. This paper is from 2015, and they analysed three methods of attack: the manipulating Tor cells method, the cell counting-based method, and the padding cell method. However, the paper is not focused on reviewing the literature.

As we have seen throughout this section, Tor has significant attention from the research community. Within Tor, hidden services is an important issue that has generated attention. However, so far no survey has analysed this issue. Therefore, a survey on HS would be useful to compile the lessons learned from its appearance and to know issues that require more research.

## 4. Systematic Literature Review Methodology

Our review of the literature of Tor hidden services which we present here is based on a systematic literature review (SLR). We chose this method to review the literature because it follows a rigorous method, unlike an expert review using ad hoc literature selection. It reduces the possibility of research bias when data selection and analysis are performed, and increases reliability, since the process could be followed by others (repeatable) [27–29]. According to [30], SLRs start by defining a review protocol that specifies the research questions that will be addressed and the methods used to develop the review. This protocol is crucial so that the review is rigorous [28,29]. Thus, an SLR is based on a defined search strategy, which must be documented, and that aims to gather as much relevant literature as possible, to which explicit inclusion and exclusion criteria are applied to evaluate each primary study. Our search process and inclusion and exclusion criteria are shown below.

In our SLR, we collected literature regarding Tor hidden services to answer the following research questions:

### 4.1. Research Questions

Our SLR on Tor HS aimed to reveal the state-of-the-art regarding HS, the research areas that are being studied, the reasons for their study, and open issues. To satisfy this goal, we defined the following research questions:

1. What are the main research areas and the main findings regarding HS?
2. What are the limitations that studies present, and in which lines of research is there not enough research regarding HS?
3. Have there been there significant advances in HS during the latest years?
4. What are the most cited articles in the area of HS?
5. Is there any relationship between launching Tor rendezvous specification version 3 and the research performed on HS?
6. What are the main future topics or problems to be investigated regarding HS?

### 4.2. Search Process

The process followed is depicted in Figure 3. Next, we explain it in detail.

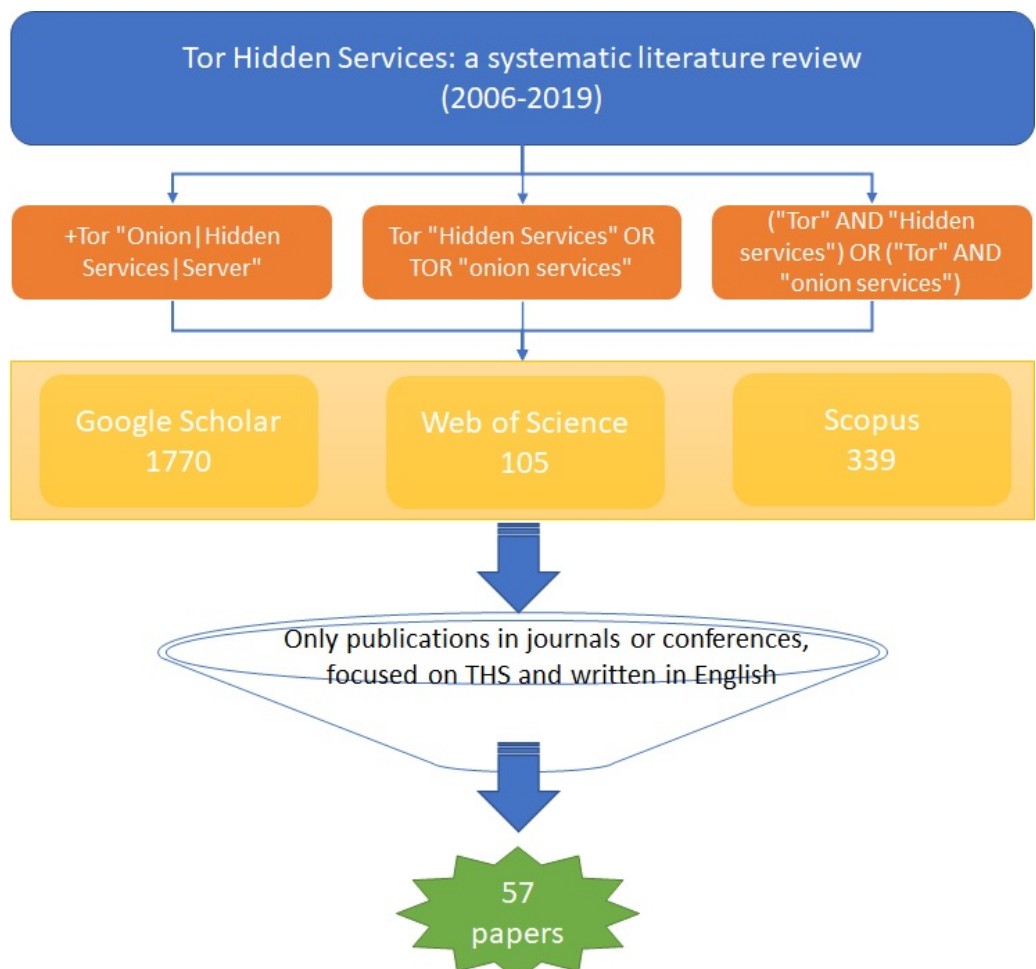

**Figure 3.** Our SLR process.

To obtain the different studies that have been used in this research, we retrieved articles from three databases: Google Scholar, Web of Science, and Scopus. We performed the query during the months of June/July 2019 using the following search terms:

- +Tor "Onion | Hidden Services | Server".
- Tor "Hidden Services" OR TOR "onion services".
- ("Tor" AND "Hidden services") OR ("Tor" AND "onion services").

These search terms were adapted to the different databases.

Taking into account that HS were born in 2004 and that what we are focused on is quite specific, we decided that would not be necessary to apply a filter regarding the years to be covered. The results we obtained after each search in the databases are shown in Table 1.

**Table 1.** Results provided by each database.

| Database | Number of Documents |
|---|---|
| Google Scholar | 1770 |
| Web of Science | 105 |
| Scopus | 339 |

*4.3. Inclusion and Exclusion Criteria*

To the results we obtained with the query in the different databases, we applied a set of inclusion and exclusion criteria to decide which studies would be analysed.

As inclusion criteria we have considered:

- The document had to be published in a journal or a conference.
- The research presented in the paper has to be focused on THS.
- The article must be written in English.

  As exclusion criteria we have considered:

- Book chapters.
- Patents.
- Citations.
- Research works that cover privacy and anonymity issues in anonymous communications systems but only cover THS in a general way.
- Technical reports.

  After applying these criteria, the number of papers was reduced to 57. The list of articles is shown in Table A1 in Appendix A.

### 4.4. Research Works and Data Analysis

Once we applied all inclusion and exclusion criteria to the papers, with the resulting collection of studies, that is, 57 papers, we gathered from each research work the following information: bibliographic information, number of references, number of citations, and type of research (review or new research work). For each paper, we also assigned a set of relevant keywords that allowed us to identify main research topics.

When we collected all this data, we analysed the papers and answered the different research questions.

## 5. Results and Discussion

Based on the analysis made of the different papers, the answers for the different research questions are provided next.

### 5.1. What Are the Main Research Areas and the Main Findings Regarding HS?

In the analysis of the selected papers on HS, we observed that they studied them from different points of view and approaches. Namely, we identified six main research areas (see Table 2): content classification; security; performance; changes in their design; discovery and measurement; and finally, human factors. In Figure 4, we can observe the classification system and the papers that contributed to more than one area. In the following sections, we analyse each area in more detail.

**Table 2.** Classification of the papers by research topics.

| Research Field | Research Papers | No. of Papers |
|----------------|-----------------|---------------|
| Content classification | [10,31–46] | 17 |
| Security | [25,26,47–72] | 28 |
| Performance | [73–76] | 4 |
| Changes in the design | [61,75,77–80] | 6 |
| Discovery and measurement | [25,32,34–36,40,44–46,76,81,82] | 12 |
| Human factors | [10,83,84] | 3 |

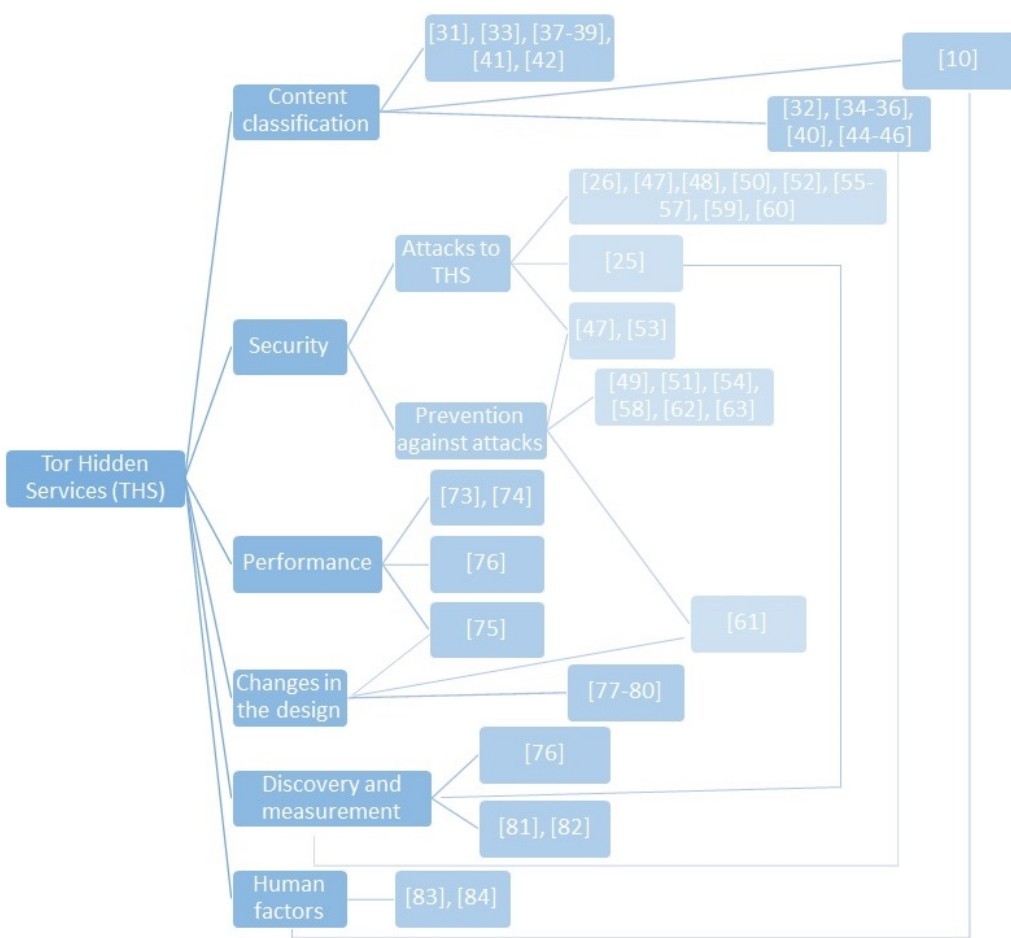

**Figure 4.** Relationships between papers and areas.

5.1.1. Security

As shown in Table 2, security is the main field of research regarding HS. Within this field, we can classify the papers into two categories. On the one hand, some articles cover specific attacks for denial-of-service (DoS) or de-anonymizing HS, and on the other hand, others are focused on preventing any of these attacks. In Table 3 we classify the papers covering security into these categories.

As we can see, most of the papers are focused on attacks. We can also point out that there are also papers that cover both possible attacks and mechanisms to prevent them as in [47,51,53] (see Figure 4). Next, we analyse the categories shown in Table 3.

**Table 3.** Classification of the papers in the security field.

| Focus of the Research | Research Papers | No. of Papers |
|---|---|---|
| Attacks to HS | [25,26,47,48,50,52,53,55–57,59,60,64–70,72] | 20 |
| Prevention against attacks | [47,49,51,53,54,58,61–63] | 9 |

Attacks to HS

As mentioned, there are two main attacks on HS: de-anonymisation attacks and DoS attacks. First, we analyse de-anonymisation attacks.

Overlier and Syverson [47] were the first to document an attack on HS. They studied the first version of the Tor hidden service protocol and claimed that one of the main vulnerabilities in Tor is the selection of the first and last nodes in the communication path.

Generally speaking, if an opponent can see the edges of a Tor circuit, then he can confirm who is communicating.

For this attack to be successful, the attacker has to control at least one node within the network, and after multiple connection attempts, he or one of these malicious onion routers (ORs) will be chosen as the first node in the rendezvous circuit established by the hidden service. From there, the attacker sends specific traffic patterns from the client to determine if one of its nodes was finally chosen as part of the circuit of interest. Once the attacker has identified that his node is the first incoming OR of the server, as shown in Figure 5, the identity of the server is revealed.

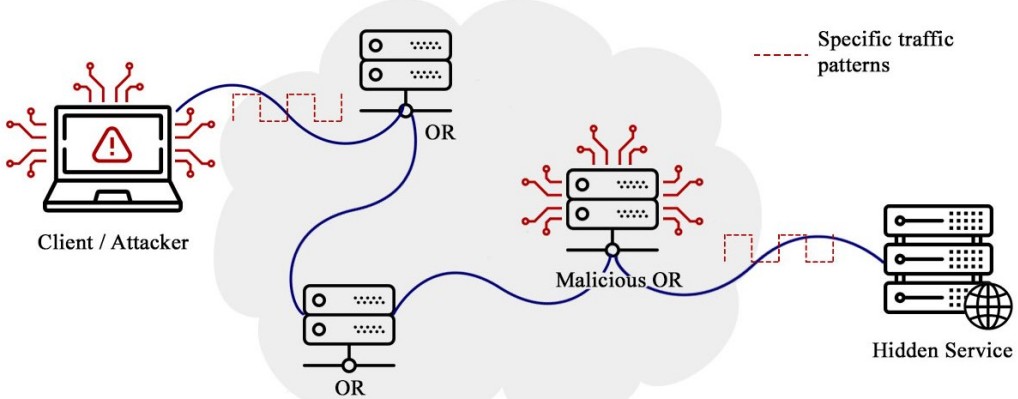

**Figure 5.** de-anonymisation THS attack.

Using the same idea of controlling the server-side input node, Zhang et al. [52] used the web browser to access the target hidden service and generate specific traffic on the circuit. If the controlled OR detects the traffic pattern, the opponent can compare and correlate the time between the web accesses and the traffic generated on the compromised router, and detect the actual IP of the server through the inbound node. Other attacks like this one based on the correlation of the traffic increase using the entry node were also described by Elices et al. [56], Ling [57], and Wang et al. [64,65].

Ma and Xu [66] presented the same idea of using an entry node and traffic flow, but in this case, to discover the identity of the client who wants to communicate with the server, assuming that this HS is malicious.

Another paper showing an attack based on the fact that it is possible to control the entry node is [71], which mentions that Tor has a TCP/Tor congestion management mechanism. This mechanism causes a data source to stop sending messages until it receives an acknowledgement of the receipt of a previous message. The researchers managed to exploit this to create an attack, by embedding a watermark on the client-side and a flow detector module on the inbound node they controlled, which monitored and analysed all the traffic passing through in order to detect the hidden server's IP.

Elices [55] and Kwon [60] demonstrated that an HS could be identified using fingerprint tracking, created from requests to the application server, disguised as a common user pattern sent by the client. In this attack, they assumed that the client could listen to the server's communication channel in order to try to detect the inserted fingerprint.

Another paper documenting a fingerprint attack [67] mentions that the larger a .onion site is, i.e., the more content it offers, the more susceptible it is to being tracked. Those that are more difficult to identify tend to be small and dynamic. Pachenko et al. [68] also analysed fingerprinting techniques and proposed their own attack. However, they concluded that neither their attack nor other existing approaches are adapted when applied in realistic environments.

Other mechanisms for de-anonymisation attacks were presented in [48,50]. The researchers exploited the fact that the frequency of clocks in a system is affected by the temperature of the CPU. The attacker via the Tor client periodically sent large amounts

of data to the hidden server, which increased its CPU temperature in such a way as to generate a clock tilt pattern, while remotely measuring changes in the clock frequencies of a set of candidate systems and trying to detect the matching pattern and thus de-anonymize the HS.

Furthermore, Matic et al. [59] discussed how information leaks in the configuration or content hosted on HS can be used to reveal their locations. They reviewed a considerable number of .onion sites for URLs or email addresses that pointed to regular websites to determine if they could be hosted on the same server, investigated HTTP certificates to extract candidate IP addresses, searched for specific HS strings, and used search engines to identify candidates hosting similar content.

The article *Trawling for Tor Services* [25] addresses several points about hidden services, including an analysis of both DoS and de-anonymisation attacks. The authors mention that if the attacker can control the access to HS descriptors, the hidden service activity can be monitored or be completely inaccessible to clients. They also highlight that attacks to discourage large-scale hidden services do not require large amounts of resources. The main vulnerabilities are described and they offer preventive measures, but these they describe as superficial.

Nepal et al. [26] reviewed the de-anonymisation attacks on hidden services raised in the literature. They analysed three kind of methods of attacks: the manipulating Tor cells method, the cell counting-based method, and the padding cell method. The articles analysed by them have also been addressed in this article. They concluded that all attack schemes need, at least, one client and one malicious guard node to de-anonymize the hidden service. Thus, in all attacks, HS are forced to choose the compromised guard nodes as their entry nodes. If this does not happen, no attempts to discover the identity of a server will be successful.

Finally, Sanatinia and Noubir [69] showed that is possible to perform a man-in-the middle (MITM) attack by compromising HS's private key (taking advantage of server's vulnerabilities or misconfigurations). The main feature of this attack is that the attacker can perform it without being in the path between the client and the server.

Next, we analyse studies on DoS attacks. Biryukov et al. [25] also asserted that DoS attacks can be performed by controlling access to HS descriptors, and they stressed that these attacks to discourage large-scale hidden services do not require large amounts of resources. Although in their research, they offered preventive measures against the main vulnerabilities found, they themselves described them as superficial.

According to these findings, some authors [70,72] pointed out that if an attacker manages to position itself as a false HSDir, it can cause an eclipse attack for denial of service. In general terms, an eclipse attack is a means of attacking decentralised networks, and then isolating and hiding a specific target, rather than attacking the entire network [70]. In the case of Tor, if an opponent can control a routing table, he/she can monopolise all the incoming and outgoing extensions of the victims, so the victim will be completely hidden. More specifically, if the first responsible directory is controlled, the attacker can monopolise or block all incoming HS nodes before customers can contact them.

Prevention against Attacks

Other related THS works have presented preventive measures to counteract the previously exposed vulnerabilities. In particular, Øverlier and Syverson [49], after exposing an attack using the entry nodes, showed that it could be counteracted by using what they call "*Valet Services*" as an extension of the concept of hidden services. The basic idea is to add an additional layer of protection for the entry points, hiding them from users. These valet nodes are now known within the Tor network as guard nodes.

Other ideas for preventing traffic analysis attacks were put forward by Beitollahi and Deconick [54], and by Yang et al. [61]. Each article proposed a different routing architecture. The first one proposed that the hidden service should also be part of the routers and that a closed circuit should be formed in the form of a ring, where data packets travel along

the circuit together with dummy packets or filler packets in both directions of the circuit. In this case, all the ORs in the ring see the traffic, but only the hidden server can understand it because only the HS can decode all the encryption layers. The second article presented a cell scheme based on multiple routes that exploits flow mixing and merging to distort or destroy the traffic patterns inserted by an attacker.

Hopper [58] conducted a study on Tor's challenges to protect hidden services from botnet abuse, which can lead to poor network performance due to increased load on nodes. Solutions to this problem include reusing failed partial circuits and isolating circuits for hidden services. For the latter case, the author stated that if a mechanism that allows ORs to recognise that a cell carries hidden service traffic is introduced, a means of protecting the rest of the system from the effects of this traffic can be provided by scheduling priority or simple isolation.

Another interesting direction is the creation of "*honey onions*" (honeypots + onion services) [62] to identify malicious directories. These Honions, as they are called in the literature, are HS whose links (.onion) are not public anywhere, which means that only the service administrators knows= about then. Therefore, when there is a possible access attempt, it is assumed to be a malicious HSDir connection attempt.

On the other hand, works such as [51,53] have approached the subject from a forensic perspective. The researchers exposed the creation of a persistent fingerprint through a flow of requests that can be recorded on a computer through a hidden service, so that, at a later time, it can be retrieved and used as evidence that a physical machine hosted particular content even after this content has been removed. This approach does not actually locate the anonymous server, but authorities can use it as almost foolproof evidence of a criminal provider's guilt.

Finally, as for MITM attacks, Sanatinia and Noubir [69] have defined a mechanism based on comparing descriptors to detect them.

### 5.1.2. Content Classification

The second topic most often addressed by researchers is the classification of content hosted or offered in hidden services. It is interesting to mention that, as far as we know, the first article that deals with this topic was written in 2013, although Tor was offering hidden services in 2004.

In this first paper on content classification [31], Guitton used three databases that listed extensive numbers of hidden services. They obtained a total of 1171 individual entries, which they then reviewed and manually classified into 23 categories. They concluded that 45% of all available HS host unethical or illegal content.

Biryukov et al. [32] also did a search for .onion addresses and as a first step found 8153 addresses, but of these, only 1813 were functional services and these were sorted into 18 categories. The results were similar to Guitton's research we explained above, since they found that 44% are dedicated to illegal services, and the remaining 56% are dedicated to different subjects that do not pose any real danger—for example, policy and anonymity forums, information resources, pages similar to WikiLeaks, and pages that provide different anonymous services, such as mail or anonymous hosting.

In addition, Biryukov et al. [32] tried to determine which the most popular Hidden services were. Their article gave the first place to an HS belonging to a zombie network structure called "Goldnet". Similarly, Biryukov et al. [25] suggested that although the list of top 10 HS included services related to BotNets, the most popular HS at that moment was Silk Road, which was shut down by the FBI in 2013. In the same way but with different results, Savage and Owen [34] found that the first place was given to HS related to abuse, although for ethical reasons they did not specify what types of abuse, nor make their .onion addresses public. The latest study, which is from 2019, shows that the most popular service is dedicated to drugs and narcotics [44]. Although none of them agreed on what is truly the most popular and influential HS, it can be inferred that the best-positioned services are those related to illicit content.

Different means of classification were presented by several authors [33,38,43]. They applied data mining techniques based on classification models to over a thousand hidden Tor services, to model their thematic organisation and linguistic diversity. Their results indicate that most of the hidden services display illegal or controversial content. In contrast, Savage and Owen [34] returned to manual study and classification, suggesting that given the variety and complex technical nature of some content, automatic classifiers would be insufficient, due to the difficulty of interpreting the context completely. Other documentation of a manual classification reviewing 3480 HS was prepared by Faizan and Khan [42], but they claimed that only 38% of the servers found offer illegal services.

On the other hand, Biswas et al. [37] presented ATOL (Automated Tool for Onion Labeling), a classification tool which includes a new discovery mechanism using keywords and a classification framework using image recognition, which was also used and replicated by Ghosh et al. [39]. Another article showing a classification mechanism using image recognition was written by Fidalgo et al. [41]. They applied improved techniques, selecting only the regions with the most outstanding information, through what is known, in machine learning, as the bag of visual words. These investigators focused on classifying services that host suspicious or illegal content, in an attempt to collaborate with entities that combat crimes involving human trafficking and child pornography.

Another scientific article showed that the vast majority of hidden services are offered in English, approximately 73.28%, and only 2.14% are offered in Spanish [40]. In the same article, the authors highlighted that there is a clear connection between the normal Internet or surface web and many hidden services. For example, they claimed that more than 20% of the domains in Tor imported resources are from the surface web and showed that approximately 90% of onion services are interconnected.

In many of the articles mentioned in this section, the authors also showed methods for the discovery of these services before classifying them. Therefore, they are mentioned below in the next category.

### 5.1.3. Discovery and Measurement

This section includes articles that retrieved as many .onion addresses as possible, and tried measuring the size of the Tor hidden network from different perspectives.

The first document is from 2009 [81], and Betzwieser et al. were the first researchers to mention that the dark network is widely interconnected. They manually located 39 hidden services, including directory sites, and from those 39, they found 20,499 more HS. Later, this concept was taken up again in 2017 by Bernaschi et al. [36], but they showed that the THS that offer specific services contain few or no references to other HS, whereas those that advertise and link to a large number of other pages are much more likely to be known and accessible from public websites.

Savage and Owen [34] attempted to discover HS by operating 40 ORs over six months, each with a bandwidth of approximately 50 kB/s and left active continuously for 25 h, with the intention that their nodes or one of them would be eligible to obtain the HSDir indicator and be able to recover the maximum number of .onion addresses possible. In the study period, they were able to observe approximately 80,000 HS. This is the first article that exposed the short life of the hidden services, many of which were observed to exist at most a few weeks before their closure. In all, only 15% of the HS they found persisted through their six-month observation period. Liu et al. [76] also took up this technique again, but with a cost evaluation, they quantified the relationship between resources consumed and hidden services collected.

Taking into account Savage and Owen's work [34] that exposed the short lifetimes of the services in Tor, Owenson et al. [82] indicated that the dark web is much smaller than many people think, since most onion services do tend to be ephemeral, so they do not all coexist at once.

Other studies [35,45] were based on the discovery of hidden services through search engines with specific keywords that allowed them to extract .onion addresses—for example,

from sites on the dark web such as wikis, or using common Surface Web search engines such as Google. Thus, Bernaschi et al. in [45] emphasised that during their collection process, they only reached 25% to 35% of the total number of hidden services that Tor says are published daily. They also pointed out that a very high percentage of these nodes are without outbound links. Furthermore, Li et al. [35] obtained 173,667 .onion addresses, but only 4,857 of these were online.

Finally, as we saw in the previous section, some articles focused on finding the most popular services on the Tor network [25,32,34,44], but most of them did so by conducting actions to discover HS and locating the most referenced .onion addresses, and the researchers who had access to HsDir [32] estimated the popularity of hidden services by looking at the request rates of descriptors.

To a lesser extent, the research analysed covered issues related to hidden services' performances and the analysis of specific parts of the protocol that they proposed improvements to. These studies are detailed below.

### 5.1.4. Performance

Loesing et al. [73] measured latency during connection to a hidden service with a special focus on general response times, and found that connection establishment when using a broadband access network took an average of 24 s. Furthermore, the study revealed that most of the time was spent establishing the connection to the introduction and rendezvous points. Lenhard et al. [74] made this same measurement but in low-bandwidth environments. They attributed the Tor bottleneck to downloading relay descriptors and building circuits. Their findings suggest an increase in the value of the timeout to avoid repeated re-transmissions.

Finally, Meng et al. [75] evaluated the performance of the hidden service domain name system they proposed, in terms of the domain name authentication, communication transmission, and domain name registration. Their results showed good performance.

### 5.1.5. Changes in the Design

Other authors present improvements or changes in Tor design that might strengthen or end the main shortcomings—for example, point to protocol changes to establish faster connections to hidden services. Their proposals include reducing the number of nodes involved in the process [47,75,80], which should lead to shorter connection establishment times.

This category also includes documents that propose changes to the circuit and data routing, specifically designed to avoid traffic recognition attacks [54,61] (see Section 5.1.1).

Meng et al. [75] proposed a new hidden service domain name system (HSDNS) that offers secure communication, increasing the anti-scanning property of the original domain name, enhancing the anti-registration attack, providing an efficient verification of authentication, and protecting users from phishing.

Finally, there is a proposal to change the .onion address from a Base32 pseudonym to a decentralized DNS that can be secure, searchable, and above all, readable [78,79].

### 5.1.6. Human Factors

In this category, we found three papers that cover issues related to human factors: illegal activities performed by means of HS and usability issues regarding them.

Regarding illegal activities taking advantage of HS, Anagnostopoulos et al. [83] showed that HS can be used to control botnets. For this, they tested the implementation of new proxy-based botnet architectures that benefit from the anonymity provided by the Tor network to disguise its command and control infrastructure. To do this, each bot creates a Tor HS, thereby acquiring an .onion address, to allow communication with the rest of the botnet. They mentioned that while this option can suffer from Tor's high latency and complex administration, it benefits greatly from stealth and anonymity, not to mention

that each HS can migrate to another physical location while maintaining the same unique onion address.

On the other hand, He et al. [10] presented a study of hidden services from a legal perspective, since HS is used by cybercriminals for the creation of services that clearly violate the laws of every country. Then, they developed a method to detect and classify the illegal activities performed on the Tor network and detect new types of illegal activities.

Finally, regarding usability, Winter et al. [84] explained how Tor users perceive, use, and administer onion services; presented what the main obstacles are for the average user; collected mental models; and obtained real usage patterns through interviews and questionnaires. Among their main findings, they mentioned that many of the people who use the services offered by Tor were not even aware that there is a significant difference from the surface web. Among other findings, they exposed the limited ways to discover the existence of onion services and usability problems, which Tor is actually improving with new versions.

### 5.2. What Are the Limitations That Research Has, and Which Lines of Research Are Lacking Regarding HS?

Very few of the investigations studied coincided with each other. There were usually many differences—for example, in the number of hidden servers found: while some discovered 20,499 [81], others managed to find approximately 80 thousand HS, with 45 thousand of these being persistent [34]. Therefore, there are many inconsistencies between the results obtained.

Another deficiency found is the lack of evidence in a realistic environment. Many of the attacks are considered successful when tested in simulated environments and are rarely brought to the real Tor network [25,43,53,55,61,64,65,80,82]. Panchenko et al. [68] presented a fingerprinting attack that is theoretically fully functional, but when brought to the real environment, had a very low recognition rate. Therefore, that the other approaches will be successful cannot be guaranteed, even if they work in simulated environments.

The problem with the research found in front of Tor's network of hidden servers is that it is a very volatile network, and the results of the studies can be confirmed one day and refuted completely the next. While many studies claim that the dark web is ruled by illegal services, others claim that the vast majority of HS host content may be sensitive in nature but is not unethical.

On the other hand, there is no study so far that validates and compares onion service discovery methods. If we compare the statistics provided by The Tor Project with the number of HS discovered, most of the scans only reached between 20% and 35% of the official total presented by Tor, and there is no evidence as to why the remaining number cannot be reached. On the other hand, until version 2 of Tor onion services, there was no significant change in the protocol that dealt with HS, and no document addresses the evolution of Tor, the increase in users, and the advancements in the mechanisms that attackers use in the face of an architecture that had evolved very little.

### 5.3. Were There Significant Advances in HS during Recent Years?

Figure 6 shows the number of research papers published by year, from the first in 2006 to the last in 2019, shortly before the completion of this work. As seen in Figure 6, it is evident that researchers have shown greater interest since 2015 began, and there was a significant increase in 2017. We can also see that there is an increasing tendency to perform more research in this field.

The biggest advancements in research in the last three years have been in the area of content classification of hidden services, such as the application of improved machine learning techniques for image recognition that allow better classification performances with smaller margins of error; crawling and .onion address discovery techniques; and to a lesser extent, security issues.

Although the techniques used present significant advances over older one, there is still no way to collect and analyse highly effective content. This issue may be due to the

slow access speed of the Tor browser, which makes it difficult to correctly observe the dynamics of hidden services.

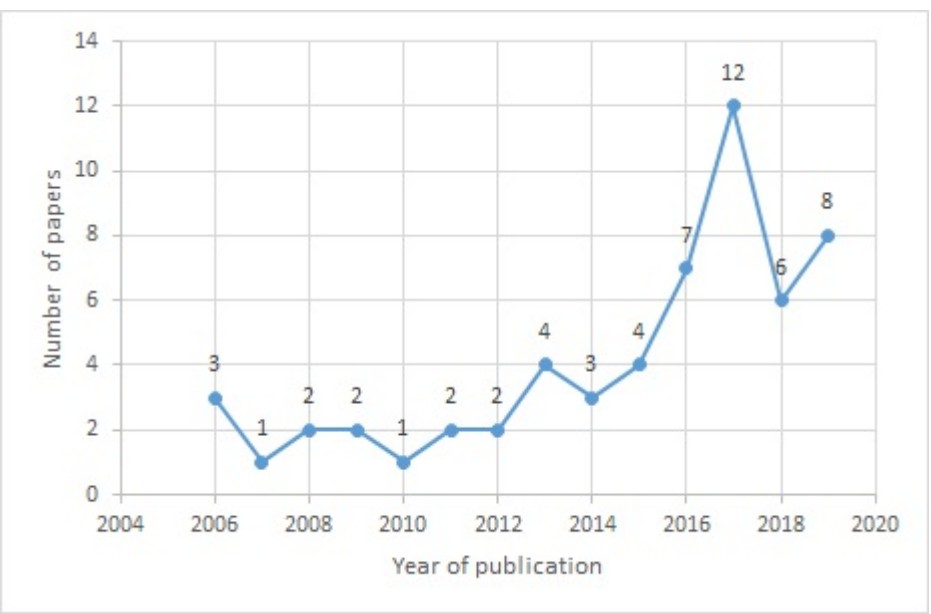

**Figure 6.** Number of publications by year.

In terms of security, the last published article explains how an eclipse attack can cause denial of service to an HS [72], but the attack mechanism still involves the control of an HSDir. Therefore, there is nothing new in comparison with other documents.

As for proposed changes to the Tor design, the articles which can be considered to cover new issues are related to changing .onion addresses to more readable domain names. However, the fact that the address assigned to each hidden service is an alphanumeric pseudo-domain has a security purpose, since it makes them difficult to trace.

### 5.4. What Are the Most Cited Articles in the Area of HS?

As mentioned in the methodology, the documents were obtained from three databases: Web of Science, Scopus, and Google Scholar. The latter is the search engine that gave the most results, including a large number of repeated results that had already been selected from the other two databases. Therefore, to homogenise the number of citations that each article received, Google Scholar was used as the source, and only in its absence were the other two used.

The three most cited articles are focused on security [25,47,48]—more specifically, on de-anonymisation. Additionally, the third one presents an analysis in terms of HS discovery. Of the articles [25,47,48], the total numbers of citations are 363, 270, and 169, respectively. It is worth mentioning that the first two were published in 2006, whereas the last one, which is the least cited, was published in 2013.

In the latter, Biryukov et al. [25] analysed the weaknesses in hidden services that can be exploited by attackers to detect, measure, and de-anonymize HS, and described several relevant attack methods, based on traffic confirmation techniques and the fact that an attacker can easily include malicious nodes in the Tor network. They also claimed that large-scale hidden service de-anonymisation attacks are practically possible with only a moderate amount of resources. They exploited Tor vulnerabilities to collect hidden service descriptors, classify their content, and detect the most popular hidden services. In short, this is one of the most comprehensive documents about Tor and its onion services.

*5.5. Is There Any Relationship between Launching Tor Rendezvous Specification Version 3 and the Research Done on HS?*

According to the Tor Project [85], in 2008, the idea of making Tor more accessible to people with less technical knowledge was introduced. Hence, they started developing the Tor Browser, a tool beyond the Tor proxy, which from 2010 positioned Tor as an instrumental tool for protecting people's identity online and granting access to critical resources, social networks, and blocked sites. The versions of the Tor protocol released have been consistent since then and to this day.

Since then, Tor has released several versions to improve performance, offer better resistance against fingerprinting attacks, correct vulnerabilities, and improve security features. In 2017, they presented the next generation of onion services at DEF CON 25, that is, the version 3. This new version introduces the following improvements: better security (cryptography, advanced client authorisation, better onion addresses, a smaller surface for targeted attacks) [85], less directory protocol leaking to directory services, and fewer offline keys for onion services. In this new generation, the onion addresses are 56 characters instead of 16 to prevent the malicious discovery of onion services. Version 2 is still working (although it is expected to stop working in Octuber 2021). Indeed, in the second half of 2020, Tor experienced the largest peak in number of .onion addresses to date [86]. Since then, this number has been decreasing progressively.

Although we expect that the deployment of the new generation of hidden services promises an increase in the number of investigations, or at least new topics to cover, none of the articles we studied included issues related to this new version.

*5.6. What Are the Main Future Subjects or Problems to Be Investigated Regarding HS?*

Hidden services are often volatile and many published services are not available [34,45,82], so researchers have an obstacle when it comes to working on discovering and measuring the actual size of the Tor network and classifying the contents of these services. Therefore, this would be a point to cover regarding the possibility of improving on or developing new mechanisms for the discovery and collection of .onion addresses. We have seen that illegal content, depending of the study, ranges from 38% to 45% [31,32,42]. One should attempt to get more realistic and updated metrics on the Tor ecosystem. At the same time, these studies should allow more accurate classifications of all services, data, and information hosted on hidden services, and indirectly, more efficient use of automated tools that try to crawl and detect content on HS, which will result in great benefits at multiple levels of research and fight against illegal content that can be easily accessed from the Tor network.

Although some advances have been made in performance, several works [73–75,80] have pointed out that this issue needs more research, since the establishment of a path with a HS is a complex and slow process. As way to improve performance, a reduction in the number of relays is proposed. At the same time, to preserve traffic analysis resistance, the use of multiple exit nodes has been proposed [80]. This is one of the issues to analyse regarding the design of Tor, but other issues, such as a HSDNS, have been proposed [75]. From our point of view, the whole design should be revised, considering the different issues proposed, in order to solve them in a global way that considers performance, security, and anonymity.

Finally, the new generation of hidden services leaves a great horizon open which must be covered from all perspectives, by addressing issues such as whether all previous vulnerabilities has been removed, whether they will be exposed to new ones, or whether the fact that new .onion addresses are more private will help the proliferation of increasingly illicit hidden services, among many other points. Finally, much of the future work should continue to address, above all, security issues, to find and eliminate vulnerabilities and security leaks.

## 6. Conclusions

Tor is currently the largest anonymous network, and the Tor protocol lends itself to in-depth research, because all its development is carried out publicly and the source code and specifications of each version are openly available.

In this article, we have presented a systematic review of the literature on Tor's hidden services. Our review showed that the dark web formed of THS is very volatile, since the HS tend to be ephemeral. Furthermore, many of the services published are not available. In this network, the HS are highly interconnected; an important percentage of these nodes have no outbound links; most of the content is in English; there is content imported from the surface web; and there is an important amount of illegal or unethical content (ranging from 38% to 45%, depending on the study). Furthermore, access to HS is slow due to the six relays that form a circuit to a HS from a client.

In this SLR, we also found that most of the papers are focused on security, either presenting attacks or proposing protection methods. Furthermore, researchers have focused on discovering at the directory level as many .onion addresses as possible, and with these addresses, classifying the content hosted on these anonymised services. To a lesser extent, research formulates changes in the protocol design to achieve faster connections to hidden services or make circuits more secure.

We also found out that many studies exposing attacks have not been tested in realistic Tor environments. Therefore, they have wide margins of failure. Another problem encountered was the inconsistency in the number of services reached or discovered using different techniques. The hidden service descriptors are stored distributively. Thus, there is no central entity storing the complete list of .onion addressed. The number of addresses found may be far from exhaustive and realistic. Therefore, it is not possible to give a size to Tor's hidden server network beyond the metrics supplied.

We have already seen that the establishment of a connection between a client a hidden service is a complex and slow process. Although research is being done on this issue, we have seen several proposals that introduced improvements regarding several issues that can improve it. We consider that this is an issue that should see more research to address it in a global and consistent way, as for other issues such as security and anonymity.

Finally, emphasis was placed on the lack of research on the new generation of hidden services. Although the new version is not the one used by most of the services deployed, according to the Tor Project, the support for version 2 of HS will stop, and it will only be possible to use version 3 of HS to give way to a much more secure ecosystem.

As future work, a deeper evaluation of the evolution of the attacks presented and their degrees of success or failure could be performed, especially since the Tor project has fixed the vulnerabilities that many of these attacks were based on.

In the medium term, this SLR should be taken up again in search of new research that has included the new generation of onion services, and we will make a comparison with the research presented here.

**Author Contributions:** Conceptualization, A.R.-M. and D.L.H.T.; Investigation, D.L.H.T. and A.R.-M.; Methodology, A.R.-M. and D.L.H.T.; Writing-original draft, A.R.-M. and D.L.H.T.; Writing-review and editing, A.R.-M. and D.L.H.T. All authors have read and agreed to the published version of the manuscript.

**Funding:** This work was funded by the European Commission's H2020 Programme under grant agreement number 830929; and the Spanish Ministry of Science, Innovation and Universities, FEDER funds, under grant numbers RTI2018-095855-B-I00 and TIN2017-86885-R.

**Institutional Review Board Statement:** Not applicable.

**Informed Consent Statement:** Not applicable.

**Data Availability Statement:** Not applicable.

**Conflicts of Interest:** The authors declare no conflict of interest.

# Appendix A

**Table A1.** List of all papers analysed.

| Number | Ref | Authors | Year |
|--------|-----|---------|------|
| 1 | [47] | Overlier and Syverson | 2006 |
| 2 | [48] | Murdoch | 2006 |
| 3 | [49] | Øverlier and Syverson | 2006 |
| 4 | [77] | Øverlier and Syverson | 2007 |
| 5 | [50] | Zander and Murdoch | 2008 |
| 6 | [73] | Loesing et al. | 2008 |
| 7 | [74] | Lenhard et al. | 2009 |
| 8 | [81] | Betzwieser et al. | 2009 |
| 9 | [51] | Shebaro et al. | 2010 |
| 10 | [52] | Lu Zhang et al. | 2011 |
| 11 | [53] | Elices et al. | 2011 |
| 12 | [54] | Beitollahi and Deconinck | 2012 |
| 13 | [55] | Elices and Perez-Gonzalez | 2012 |
| 14 | [25] | Biryukov et al. | 2013 |
| 15 | [31] | Guitton | 2013 |
| 16 | [56] | Elices and Perez-Gonzalez | 2013 |
| 17 | [57] | Ling et al. | 2013 |
| 18 | [32] | Biryukov et al. | 2014 |
| 19 | [33] | Spitters et al. | 2014 |
| 20 | [58] | Hopper | 2014 |
| 21 | [26] | Nepal et al. | 2015 |
| 22 | [59] | Matic et al. | 2015 |
| 23 | [61] | Yang and Li | 2015 |
| 24 | [60] | Kwon et al. | 2015 |
| 25 | [34] | Savage and Owen | 2016 |
| 26 | [35] | Li et al. | 2016 |
| 27 | [62] | Sanatinia and Noubir | 2016 |
| 28 | [63] | Nurmi et al. | 2016 |
| 29 | [64] | Wang et al. | 2016 |
| 30 | [65] | Wang et al. | 2016 |
| 31 | [78] | Victors et al. | 2016 |
| 32 | [36] | Bernaschi et al. | 2017 |
| 33 | [37] | Biswas et al. | 2017 |
| 34 | [38] | Nabki et al. | 2017 |
| 35 | [39] | Ghosh et al. | 2017 |
| 36 | [40] | Sanchez-Rola et al. | 2017 |
| 37 | [66] | Ma and Xu | 2017 |
| 38 | [67] | Overdorf et al. | 2017 |
| 39 | [68] | Panchenko et al. | 2017 |
| 40 | [69] | Sanatinia and Noubir | 2017 |
| 41 | [75] | Meng et al. | 2017 |
| 42 | [83] | Anagnostopoulos et al. | 2017 |
| 43 | [70] | Tan et al. | 2017 |
| 44 | [71] | Iacovazzi et al. | 2018 |
| 45 | [76] | Liu et al. | 2018 |
| 46 | [79] | Meng et al. | 2018 |
| 47 | [84] | Winter et al. | 2018 |
| 48 | [80] | Liang and Liu | 2018 |
| 49 | [82] | Owenson et al. | 2018 |
| 50 | [10] | He et al. | 2019 |
| 51 | [41] | Fidalgo et al. | 2019 |
| 52 | [42] | Faizan and Khan | 2019 |
| 53 | [43] | Takaaki and Atsuo | 2019 |
| 54 | [44] | Al-Nabki et al. | 2019 |
| 55 | [45] | Bernaschi et al. | 2019 |
| 56 | [46] | Park et al. | 2019 |
| 57 | [72] | Tan et al. | 2019 |

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
