# Peer review of "Tor Hidden Services: A Systematic Literature Review"

_jcp, doi:10.3390/jcp1030025_

Round 1

Reviewer 1 Report

Dear authors,

Hereafter all my comments about your research paper:

Line 4: Please be more clear in the abstract saying "we can mention user’s IP location or Tor Hidden Services (THS) as a mechanism to conceal the location of servers, mainly, web servers".
Line 6: Fix "sump" typo.
Line 24: Please explain what you mean for "anonymous services".
Line 27: Fix "According to".
Line 30: Please clarify if you are referring to THS or simply to the anonymous navigation offered by the Tor network.
Line 50: Do you mean "services" or "features"?
Lines 70-74: Please clarify the number of nodes required and why. It is not clear.
Lines 75-88: Please add a scheme to explain the protocol.
Line 101: Fix "vulnerabilities" to "vulnerability".
Lines 130-131: Please explain "flow watermarking attacks" or refer to other parts in the paper.
Lines 209-210: Please be more explicit about the "others" category since only 2 papers are included in this category.
Line 232: OR acronym is not specified.
Section 5.1.1: Please do not mix DoS and deanonymizazion attacks. Try to analyze them in order.
Section 5.1.1: It would be easier to understand attacks by adding a drawing (maybe just for the simplest deanonymizazion attack).
Line 390: Please clarify MSM.
Section 5.1.3: You have mixed in this section something related to content classification (previous section).
Section 5.1.4: Please better explain why [66] has been placed in this category.
Line 518: Google Scholar is not a good way to track paper citations.
Section 5.5: I don't understand if you want to show correlation or not between research and Tor updates.

The overall review approach seems good, but I think you should add at least a section with much more technical details about the protocol. You have spent sections to described how you found papers, but you missed several technical details about Tor.

Author Response

We provide our response both here and as a PDF file.

Dear Editor and Reviewers,

First of all, we would really like to thank the reviewers and editors for all the time and effort spent on providing their feedback, suggestions and comments, which have helped us a lot to improve our manuscript. Second, considering time limitations for the review, we have tried to do our best to address reviewers' comments, doubts and concerns related to our manuscript. Below we outline the main improvements that we have performed on the new version:

  • We have carefully reviewed the text to remove typos and improve English language and the writing in all the sections of our paper for the sake of clarity and completeness.
  • The abstract has been improved to clarify some statements that were not clear.
  • The introduction has been changed to improve the explanation of Tor Hidden Services.
  • Section 2, Tor Hidden Services in a nutshell provides a more detailed explanation on how the protocol works. New figures have been added for this purpose.
  • We have improved the classification of the papers and we have renamed the “Others” group to “Human factors”.
  • Section 5.1.1 the part of attacks section has been reorganized to explain, first, deanonymization attacks and then, DoS attacks. We have also added a figure to show the deanonymization attack.
  • We have revised Section 5.1.3 regarding content classification and explained its relationship with Section 5.1.2.
  • The research question in Section 5.5 has been reformulated to focus it on THS.
  • We have improved Section 5.6 and conclusions to show the main issues that deserve more research.
  • We have completed the information of some references that were incomplete, and we have added new ones to explain and justify additional the issues required by reviewers.

Thank you for considering our work.

Best regards,

Diana L. Huete Trujillo, and Antonio Ruiz-Martínez          

Reviewer #1’s Comments

[Reviewer #1] Dear authors, Hereafter all my comments about your research paper:

[Authors] We would like to first thank Reviewer 1 for his/her comments and constructive critics. They have helped us to improve and enhance our work. We really appreciate his/her efforts in deeply reviewing our paper.

[Reviewer #1] Line 4: Please be more clear in the abstract saying "we can mention user’s IP location or Tor Hidden Services (THS) as a mechanism to conceal the location of servers, mainly, web servers".

[Authors] We have modified the abstract so that it be clearer.

[Reviewer #1] Line 6: Fix "sump" typo. Line 27: Fix "According to".

[Authors] We have fixed all the typos you mentioned. Additionally, we have revised carefully all the document to correct other typos and improve English language.

[Reviewer #1] Line 24: Please explain what you mean for "anonymous services".

[Authors] We have rewritten the text to explain clearly what hidden or onion services offer.

[Reviewer #1] Line 30: Please clarify if you are referring to THS or simply to the anonymous navigation offered by the Tor network.

[Authors] We have clarified this issue by improving the text that refers to the hidden services and the citizen accessing the service.

[Reviewer #1] Line 50: Do you mean "services" or "features"?

[Authors] We meant services. Namely, hidden services. We have clarified it in the text.

[Reviewer #1] Lines 70-74: Please clarify the number of nodes required and why. It is not clear.

[Authors] We have rewritten the text to clarify that the connection between the user a hidden service requires to build 2 Tor circuits. As the establishment of each circuit requires the use of 3 onion routers, the whole connection requires, therefore, 6 nodes.

[Reviewer #1] Lines 75-88: Please add a scheme to explain the protocol.

[Authors] Following your recommendation, we have added a figure where the different steps that are followed during the protocol. We have also changed the text to explain it in more detail according to the Figure shown.

[Reviewer #1] Line 101: Fix "vulnerabilities" to "vulnerability".

[Authors] We have fixed it, thanks.

[Reviewer #1] Lines 130-131: Please explain "flow watermarking attacks" or refer to other parts in the paper.

[Authors] We have included a short description of what a flow watermarking attack is.

[Reviewer #1] Lines 209-210: Please be more explicit about the "others" category since only 2 papers are included in this category.

[Authors] We have been made more explicit and we have changed the name of the category from “Others” to “Human factors” since these two papers are related to these issues. We have made all the changes necessaries throughout the paper to reflect this change.

[Reviewer #1] Line 232: OR acronym is not specified.

[Authors] We have put what it stands for. In Section 5.1.1., we have also added a figure to show the deanonymization attack.

[Reviewer #1] Section 5.1.1: Please do not mix DoS and deanonymizazion attacks. Try to analyze them in order.

[Authors] Following your recommendation, we have reorganized to explain, first, deanonymization attacks and then, DoS attacks. We have also added a figure to show the deanonymization attack.

[Reviewer #1] Section 5.1.1: It would be easier to understand attacks by adding a drawing (maybe just for the simplest deanonymizazion attack).

[Authors] We have added a figure to show the deanonymization attack, as you suggested.

[Reviewer #1] Line 390: Please clarify MSM.

[Authors] We made a mistake; we have corrected it since we refer to hidden services.

[Reviewer #1] Section 5.1.3: You have mixed in this section something related to content classification (previous section).

[Authors] We have revised Section 5.1.3 regarding content classification and explained its relationship with the previous section.

[Reviewer #1] Section 5.1.4: Please better explain why [66] has been placed in this category.

[Authors] Thanks for your comment, we realised that we marked incorrectly the paper. We have updated the text and the classification to put it in the Human factor group.

[Reviewer #1] Line 518: Google Scholar is not a good way to track paper citations.

[Authors] Although there was some publication that indicates that Google Scholar (GS) citations can be manipulated [1]. There are several publications that indicate that GS “GS provides a more nuanced and comprehensive representation of research impact and international scope than the commercial databases.” [2] and that “is essentially a superset of WoS and Scopus, with substantial extra coverage” [3] and “the wide document coverage of Google Scholar (specially books and book chapters) enables more comprehensive analyses of the documents published in a specific discipline than were previously possible with other citation indexes” [4]. Furthermore, “the use of citations reported in GS is appropriate for evaluating research impact in disciplines where other formats beyond the English-language journal article are valued” [4] and “Spearman correlations between citation counts in GS and WoS or Scopus are high (0.78-0.99).” [3]. Therefore, we consider that GS could be used as a measure the impact of a publication.

[1] E. D. Lopez-Cozar, N. Robinson-Garcia, and D. Torres-Salinas, «Manipulating Google Scholar Citations and Google Scholar Metrics: simple, easy and tempting», arXiv:1212.0638 [cs], feb. 2013, Accessed: ago. 24, 2021. [On line]. Available in: http://arxiv.org/abs/1212.0638

[2] K. Chapman and A. E. Ellinger, «An evaluation of Web of Science, Scopus and Google Scholar citations in operations management», The International Journal of Logistics Management, vol. 30, n.º 4, pp. 1039-1053, ene. 2019, doi: 10.1108/IJLM-04-2019-0110.

[3] A. Martín-Martín, E. Orduna-Malea, M. Thelwall, and E. Delgado López-Cózar, «Google Scholar, Web of Science, and Scopus: A systematic comparison of citations in 252 subject categories», Journal of Informetrics, vol. 12, n.º 4, pp. 1160-1177, nov. 2018, doi: 10.1016/j.joi.2018.09.002.

[4] A. Martín-Martín, E. Orduna-Malea, and E. Delgado López-Cózar, «A novel method for depicting academic disciplines through Google Scholar Citations: The case of Bibliometrics», Scientometrics, vol. 114, n.º 3, pp. 1251-1273, mar. 2018, doi: 10.1007/s11192-017-2587-4.

[Reviewer #1] Section 5.5: I don't understand if you want to show correlation or not between research and Tor updates.

[Authors] Our main aim was to show if there was a correlation between Tor updates regarding hidden services and research. However, we were analysing all updates regarding Tor, which are out of the scope of our paper. Then, we have redefined the research question to focus on Hidden services, which is the main purpose of this paper, and we have rewritten this section.

[Reviewer #1] The overall review approach seems good, but I think you should add at least a section with much more technical details about the protocol. You have spent sections to described how you found papers, but you missed several technical details about Tor.

[Authors] Following your recommendation, we have improved Section 2 to provide more details and we have included some images to explain the process.

Reviewer 2 Report

The authors of "Tor Hidden Services: a systematic literature review" aim to provide an overview of existing research to date on Tor and, in particular, Tor Hidden Services. The paper is written fairly clearly (although a close edit to correct small errors is needed) and the description of the methods sounds convincing. However, as a scholar working on this subject myself, I did not learn much from the paper and was shocked to find several of the leading researchers in this area absent.

Eric Jardine's important work on the Tor Dark Web as well as Tor's potential for political freedom is missing. So is Nicolas Christin's work on cryptomarkets which is crucial to understanding THS. There are no references whatsoever to the journal Surveillance & Society, which has dozens of relevant studies. In short, even though the methods sound good, they have not captured much of the existing literature.

More importantly, the paper does not offer much an original synthesis that I wouldn't get by reading almost any existing single empirical paper. I would encourage the authors to be more analytical in identifying where the research ought to go.

I wish the authors the best of luck in revising this paper. I think it has promise, but it simply must have better coverage of the literature.

Author Response

We provide our responses both here and as a PDF file.

Dear Editor and Reviewers,

First of all, we would really like to thank the reviewers and editors for all the time and effort spent on providing their feedback, suggestions and comments, which have helped us a lot to improve our manuscript. Second, considering time limitations for the review, we have tried to do our best to address reviewers' comments, doubts and concerns related to our manuscript. Below we outline the main improvements that we have performed on the new version:

  • We have carefully reviewed the text to remove typos and improve English language and the writing in all the sections of our paper for the sake of clarity and completeness.
  • The abstract has been improved to clarify some statements that were not clear.
  • The introduction has been changed to improve the explanation of Tor Hidden Services.
  • Section 2, Tor Hidden Services in a nutshell provides a more detailed explanation on how the protocol works. New figures have been added for this purpose.
  • We have improved the classification of the papers and we have renamed the “Others” group to “Human factors”.
  • Section 5.1.1 the part of attacks section has been reorganized to explain, first, deanonymization attacks and then, DoS attacks. We have also added a figure to show the deanonymization attack.
  • We have revised Section 5.1.3 regarding content classification and explained its relationship with Section 5.1.2.
  • The research question in Section 5.5 has been reformulated to focus it on THS.
  • We have improved Section 5.6 and conclusions to show the main issues that deserve more research.
  • We have completed the information of some references that were incomplete, and we have added new ones to explain and justify additional the issues required by reviewers.

Thank you for considering our work.

Best regards,

Diana L. Huete Trujillo, and Antonio Ruiz-Martínez          

Reviewer #1’s Comments

[Reviewer #2] The authors of "Tor Hidden Services: a systematic literature review" aim to provide an overview of existing research to date on Tor and, in particular, Tor Hidden Services. The paper is written fairly clearly (although a close edit to correct small errors is needed) and the description of the methods sounds convincing.

[Authors] We would like to first thank Reviewer #2 for his/her comments and constructive critics. They have helped us to improve and enhance our work. We really appreciate his/her efforts in reviewing our paper.

[Reviewer #2] However, as a scholar working on this subject myself, I did not learn much from the paper and was shocked to find several of the leading researchers in this area absent. Eric Jardine's important work on the Tor Dark Web as well as Tor's potential for political freedom is missing. So is Nicolas Christin's work on cryptomarkets which is crucial to understanding THS. There are no references whatsoever to the journal Surveillance & Society, which has dozens of relevant studies. In short, even though the methods sound good, they have not captured much of the existing literature.

[Authors] Thanks for your comments and for your suggestion as for important references in the literature. It is important to point out that for doing our literature review, we have followed a systematic literature review. Unlike to a usual review, where the selection process is made ad-hoc and guided by researchers’ criteria or point of view, a systematic literature review follows a rigorous methodological process based on a defined search strategy aiming at being valid, reliable, repeatable, and reducing researcher bias in data selection and analysis. In this revised version of the paper, we have included this justification in Section 4.

Then, our selection of the literature is based on the procedure we presented in Section 4. In this section, we presented the search process, which is conducted by a set of key words that have been used to search in different databases. Next, we have applied inclusion and exclusion criteria. This provided us the set of papers to be analysed. If you see our keywords and our process you can see that our review is focused on hidden services and then, the papers provided by the databases are those that are analysing different issues regarding hidden services. The papers you suggested us are at a high level of abstraction, since they are focused on the dark web. The dark web is created from hidden services. However, the papers you suggested are not analysing them in a specific issue, they are covering its usage from a high-level perspective, and this is the reason why, probably, they have not appeared in the queries we launched. Even though, as we consider that you provided us important references, we have included them in the introduction. As you did not provide us the whole reference, we have tried to locate the papers in based on the author’s name and the topic you mentioned. From our point of view, the references you aim at indicating us are the following ones, we hope we have selected accurately.

Jardine, Eric, The Dark Web Dilemma: Tor, Anonymity and Online Policing (September 30, 2015). Global Commission on Internet Governance Paper Series, No. 21, Available at SSRN: https://ssrn.com/abstract=2667711 or http://dx.doi.org/10.2139/ssrn.2667711

Martin, J., & Christin, N. (2016). Ethics in cryptomarket research. International Journal of Drug Policy, 35, 84-91. https://doi.org/10.1016/j.drugpo.2016.05.006

[Reviewer #2] More importantly, the paper does not offer much an original synthesis that I wouldn't get by reading almost any existing single empirical paper. I would encourage the authors to be more analytical in identifying where the research ought to go.

[Authors] In general, a single empirical paper on Tor Hidden Services, in the related work, does not either cover the analysis of 57 papers nor shows the main findings and issues to be addressed as this review does. However, probably, we have not properly expressed the future research issues. We have made changes in Section 5.6 and in conclusions to try to address this issue.

[Reviewer #2] I wish the authors the best of luck in revising this paper. I think it has promise, but it simply must have better coverage of the literature.

[Authors] Considering your comments, reviewer’s #1 comments, and the time the editor gives us, we have done our best to improve the paper. As for the review of the literature, as we explained previously, we have reviewed it according to the protocol we defined following the SRL method.

Round 2

Reviewer 1 Report

I really appreciate the extensive work that has been done to implement my comments. The paper has improved very much and it is also more clear thanks to the inclusion of new figures and concepts. The paper now deserves to be published in JCP.

Reviewer 2 Report

The authors are to be commended on making some major improvements in a short period of time. Sections 4 and 5.6 especially are much stronger. I am persuaded by the authors' argument for the contribution of their paper.

I understand the point the authors are making about how a "systematic" lit review differs from an expert lit review. It *reduces* bias. But there is still bias in the selection of search terms. My original point was that if important and relevant articles are excluded given the systematic methods that suggests a problem of face validity (i.e., maybe the selected search terms should be different). The authors have accurately carried out the study given these search terms. But what does strike me is that this paper reveals is that computational social scientists are using very different language than computer scientists, infosec researchers, and engineers to describe the same things. That produces a problem that for interdisciplinary dialogue that is well beyond the scope of the authors' work.

In short, I think the authors succeed in offering a systematic lit review for the search terms they selected. It provides a useful overview of some findings through summer 2019 on THS.